# Correlation between histogram-based DCE-MRI parameters and $^{18}$F-FDG PET values in oropharyngeal squamous cell carcinoma: Evaluation in primary tumors and metastatic nodes

Antonello Vidiri[1], Emma Gangemi[1,2]*, Emanuela Ruberto[1], Rosella Pasqualoni[3], Rosa Sciuto[3], Giuseppe Sanguineti[4], Alessia Farneti[4], Maria Benevolo[5], Francesca Rollo[5], Francesca Sperati[6], Filomena Spasiano[4], Raul Pellini[7], Simona Marzi[8]

1 Radiology and Diagnostic Imaging Department, IRCCS Regina Elena National Cancer Institute, Rome, Italy, 2 Departmental Faculty of Medicine and Surgery, Center for Integrated Research, University Campus Bio-Medico of Rome, Rome, Italy, 3 Department of Nuclear Medicine, IRCCS Regina Elena National Cancer Institute, Rome, Italy, 4 Department of Radiotherapy, IRCCS Regina Elena National Cancer Institute, Rome, Italy, 5 Department of Pathology, IRCCS Regina Elena National Cancer Institute, Rome, Italy, 6 Biostatistics-Scientific Direction, IRCCS Regina Elena National Cancer Institute, Rome, Italy, 7 Department of Otolaryngology & Head and Neck Surgery, IRCCS Regina Elena National Cancer Institute, Rome, Italy, 8 Medical Physics Laboratory, IRCCS Regina Elena National Cancer Institute, Rome, Italy

* emma_gan86@yahoo.it

## Abstract

### Objectives

To investigate the correlation between histogram-based Dynamic Contrast-Enhanced magnetic resonance imaging (DCE-MRI) parameters and positron emission tomography with $^{18}$F-fluorodeoxyglucose ($^{18}$F-FDG-PET) values in oropharyngeal squamous cell carcinoma (OPSCC), both in primary tumors (PTs) and in metastatic lymph nodes (LNs).

### Methods

52 patients with a new pathologically-confirmed OPSCC were included in the present retrospective cohort study. Imaging including DCE-MRI and $^{18}$F-FDG PET/CT scans were acquired in all patients. Both PTs and the largest LN, if present, were volumetrically contoured. Quantitative parameters, including the transfer constants, $K^{trans}$ and $K_{ep}$, and the volume of extravascular extracellular space, $v_e$, were calculated from DCE-MRI. The percentiles (P), P10, P25, P50, P75, P90, and skewness, kurtosis and entropy were obtained from the histogram-based analysis of each perfusion parameter. Standardized uptake values (SUV), $SUV_{max}$, $SUV_{peak}$, $SUV_{mean}$, metabolic tumor volume (MTV) and total lesion glycolysis (TLG) were calculated applying a SUV threshold of 40%. The correlations between all variables were investigated with the Spearman-rank correlation test. To exclude false positive results under multiple testing, the Benjamini-Hockberg procedure was applied.

**Data Availability Statement:** All relevant data are within the paper and its Supporting Information files.

**Funding:** This work was supported by the Italian Association for Cancer Research (AIRC, project No.17028).

**Competing interests:** The authors have declared that no competing interests exist.

## Results

No significant correlations were found between any parameters in PTs, while significant associations emerged between $K^{trans}$ and $^{18}$F-FDG PET parameters in LNs.

## Conclusions

Evident relationships emerged between DCE-MRI and $^{18}$F-FDG PET parameters in OPSCC LNs, while no association was found in PTs. The complex relationships between perfusion and metabolic biomarkers should be interpreted separately for primary tumors and lymph-nodes. A multiparametric approach to analyze PTs and LNs before treatment is advisable in head and neck squamous cell carcinoma (HNSCC).

## Introduction

Head and neck squamous cell carcinoma (HNSCC) is the sixth most common cancer world-wide [1]. In the last few decades, there has been an increase in the incidence of oropharyngeal squamous cell carcinoma (OPSCC) related to human papilloma virus (HPV), a distinct entity from the traditional tobacco- and alcohol-related OPSCC [2].

Magnetic resonance imaging (MRI) and positron emission tomography with $^{18}$F-fluoro-deoxyglucose ($^{18}$F-FDG-PET) are the current diagnostic imaging methods for staging and treatment monitoring of HNSCC [3–5]. In recent years, dynamic contrast-enhanced MRI (DCE-MRI) and diffusion-weighted imaging (DWI) have also been introduced in clinical practice to obtain a more comprehensive characterization of HNSCC, based on functional parameters related to tissue microvascular properties and cellularity, respectively [6].

Concurrently, some histopathological parameters, such as p16 expression and proliferation index, measured from KI67 labelling, have been proposed to predict the tumor behaviour in HNSCC, as they can provide information about tumor aggressiveness, prognosis, and therapy response [7–8]. Other biomarkers, such as epidermal growth factor receptor (EGFR) and tumor suppressor protein p53 expression, have been investigated for their potential capability to support personalized treatment protocols, enabling the categorization of patients into different risk groups [9].

Considering the volume of data that can be derived from both histology and functional imaging, radiologists and clinicians should be aware of the different potential roles of several biomarkers with respect to specific clinical end points. To this purpose, a number of reports have recently evaluated the complementarity and/or associations between imaging and histo-pathological features in HNSCC [10–15], as well as in different malignancies, i.e. breast cancer, lung adenocarcinoma and glioma [16–18]. The ultimate goal of this is to determine which parameters or their combinations could be appropriately used in clinical practice for a more precise diagnosis and treatment of these cancers. It was found that the apparent diffusion coefficient (ADC) is able to predict cell count and proliferation activity, while although $SUV_{max}$ may predict expression of HIF-1α, it is not a good surrogate marker for KI67 labelling and p53 expression [10]. DCE-MRI parameters also were demonstrated to be related to different histo-pathological features, such as vessel count, total vessel area [11] and microvessel density [15].

A better understanding of the complex interactions between functional imaging parameters is also advisable, as it may expand our knowledge of tumor biological characteristics, with potential clinical implications for treatment planning, prediction of treatment response and

patient outcome. Several investigations have already focused on the relationships between DCE-MRI and [18]F-FDG-PET/computed tomography (CT) parameters, even though these results are conflicting in HN tumors [19–25]. Most of the previous studies on DCE-MRI and/ or PET/CT have only evaluated primary tumors [19–21,23,24,26], while only a small number of investigations also included the metastatic lymph nodes (LNs) [25,27–28]. Furthermore, limited research has addressed vascular heterogeneity within the lesion, using a histogram-based approach instead of the mean values of parameters, to better reveal the relationships between perfusion and metabolic variables [19,22].

Thus, the aim of our study was to further investigate the relationships between DCE-MRI and [18]F-FDG-PET/CT parameters in OPSCC. To our knowledge, this is the first study investigating the correlation between histogram-based analysis of DCE-MRI parameters and volumetric [18]F-FDG-PET values in a large and homogeneous population of OPSCC, both in primary tumors (PTs) and in metastatic LNs.

## Materials and methods

### Patient population

This cohort study was conducted at the IRCCS Regina Elena National Cancer Institute, Rome, Italy. It was conducted retrospectively on a patient population that was also included in a larger prospective study funded by the Italian Association for Cancer Research (project No. 17028) in OPSCC, aiming to investigate the ability of DCE-MRI and DWI to predict tumor response to chemo-radiotherapy.

The study was authorized by the hospital ethics committee i.e. 'Central Ethics Committee, IRCCS LAZIO, IFO' with a reference number of N1214/19. Patient records have been anonymized at the end of the study to create an anonymous database, which has been provided as Supporting Information file. Due to the retrospective nature of the study and the lack of published data that could have supported a specific hypothesis for a conventional sample size calculation, we considered a sample size of 50 patients as adequate, based on the number of patients coming into our institute in the observational period selected.

Inclusion criteria were: (i) aged 18 years or older; (ii) Karnofsky performance status > 80; (iii) pathologically confirmed OPSCC; (iv) stage III or IV without distant metastases according to the 8[th] edition of American Joint Committee on Cancer (AJCC) staging system; (v) treatment with radiotherapy ± chemotherapy; (vi) DCE-MRI and [18]F-FDG PET/CT performed at our institute during diagnosis. Exclusion criteria were: (i) any contraindication to MR examination; (ii) the presence of artifacts in the images that do not allow a quantitative evaluation; (iii) prior surgery or chemoradiotherapy to the primary disease and the neck. Specific informed consent was obtained from each patient.

All patients' tissue samples and medical records were accessed between November 2018 and April 2019. Demographic data of the enrolled patients were obtained and tumor subsites were recorded. T and N classifications were (re)staged according to the 8th edition of AJCC staging system.

### HPV testing

HPV-positive OPSCCs were identified by using both p16 immunohistochemistry and PCR-based detection techniques. HPV-positive patients were defined as those with both p16 and HPV-DNA positivity [29].

Formalin-fixed paraffin-embedded (FFPE) tissue was obtained from patients and each block was sectioned into 1–3 x 5 μm slices, depending on the tissue size available. DNA was purified using the DNeasy Blood and Tissue Kit (Qiagen). The PCR-based INNO-LiPA HPV

Genotyping Extra II kit (Fujirebio) and TENDIGO™ instrument (Fujirebio) were used to detect and genotype HPV-DNA. This assay allows the identification of 32 high risk and low risk HPV types.

The p16 protein expression was assessed using the CINtec® Histology Kit (Roche Diagnostics, Milan, Italy). The staining was evaluated according to the AJCC (American Joint Committee on Cancer) Staging Manual, 8th Edition.

Histological grading of OPSCC was described according to the AJCC Staging Manual. Specifically, histological grading has been applied only for the HPV-negative OPSCCs, as no grading system currently exists for HPV-positive OPSCCs [30].

## MR imaging protocol

The MRI exams were acquired with a 1.5-T system (Optima MR 450w, GE Healthcare, Milwaukee, WI) with 16-channels receive-only RF coils: a head, a surface neck, and a spine coil.

The MRI protocol included coronal fast spin-eco (FSE) T2-weighted images (acquisition matrix: 288×256, field of view: 27×27 cm, TR/TE: 5901/102 ms; slice thickness: 4 mm), axial FSE T2-weighted images (acquisition matrix: 288×256, field of view: 20×20 cm, TR/TE: 6844/105 ms; slice thickness: 3 mm), and pre-contrast axial T1-weighted images (acquisition matrix: 288×256, field of view: 20×20 cm, TR/TE: 617/8.1 ms; slice thickness: 3 mm), all acquired from the skull base to the level of the thoracic inlet. Axial DWI was obtained via single-shot spin-echo and echo-planar imaging (acquisition matrix: 128×128; field of view: 26×28 cm; TR/TE: 4500/77 ms; slice thickness: 3 mm, b value: 0-500-1000). DCE-MRI involved a 3D fast-spoiled gradient echo sequence, with a TR/TE of 4.9/1.60 ms, flip angle 30˚, acquisition matrix 128X128, field of view 28 cm, number of slices 20, slice thickness of 4 mm, no spacing. Sixty dynamic volumes were acquired consecutively, with a temporal resolution of 5 s, and a total scanning time of 5 min and 15 s. At the fourth dynamic volume, 0.1 mmol/kg body weight of gadopentetate dimeglumine contrast agent was administered intravenously, at a rate of 3 ml/s. After contrast administration, axial and coronal T1-weighted images with liver acquisition with volume acceleration sequences were acquired (LAVA; acquisition matrix: 288×288, field of view: 26×26 cm, TR/TE 9.8/3 ms; slice thickness: 1 mm, acquisition time of 2.05 min).

## [18]F-FDG-PET /CT image acquisition

Combined PET/CT imaging was performed using a non-TOF (Time of Flight) tomography (Biograph 16, Siemens). All patients fasted for at least 6 hours prior and were preconditioned to have a blood glucose level <150 mg/dl at the time of injection of FDG. [18]F-FDG-PET/CT acquisition was performed 60±10 min. after intravenous (i.v.) injection of an average dose of 5 MBq/Kg of [18]F-FDG. A non-contrast enhanced CT scan from the base of the skull to the upper thighs was acquired for anatomical localisation and attenuation correction of PET images, with the following parameters: 120–140 kV, 4 mm slice thickness. PET data were acquired in 3D mode immediately after the CT scan, taken for 2–3 minutes at each bed position. PET images were reconstructed by the TrueX algorithm, that employs a system matrix with point spread function modelling, with three iterations and 21 subsets. After reconstruction the images were filtered by a Gaussian filter with a full width at half maximum of 4 mm. PET images were finally corrected for attenuation using data from the CT scan.

## DCE-MRI analysis and tumor delineation

A commercial software package (GenIQ General, GE Advanced Workstation, Palo Alto, CA) was used to analyze the DCE-MRI data. A pharmacokinetic modeling based on two compartments (plasma space and extravascular-extracellular space) was applied to obtain the following

quantitative parameters: $K^{trans}$, the transfer constant between plasma and the extravascular extracellular space (EES), $K_{ep}$, the transfer constant between EES and plasma and $v_e$, the fractional volume of EES [31]. MIM software (v.6.4.2, MIM Software Inc., USA) was used to visualize axial T2-weighted images and manually delineate the volume of the PT and the largest metastatic LN, if present, by an expert HN radiologist with more than 20 years of experience (A.V.). Arterial or venous structures, bony components and macroscopic necrosis were excluded from the lesions. The lesion contours, as well as the perfusion maps of $K^{trans}$, $K_{ep}$ and $v_e$ were uploaded to the Matlab workspace (Release 7.10.0, The Mathworks Inc., Natick, MA), where dedicated scripts were developed for subsequent quantitative analyses. From the volumetric histogram of each perfusion parameter, the following eight variables were calculated: skewness, kurtosis, and entropy, as well as the $10^{th}$, $25^{th}$, $50^{th}$ (median value), $75^{th}$ and $90^{th}$ percentiles.

The same bin size was used for each patient to calculate the histogram distribution of the parameters within the lesion; in particular, the bin sizes were 0.05 $min^{-1}$, 0.3 $min^{-1}$, and 0.02 for $K^{trans}$, $K_{ep}$, and $v_e$, respectively. The volume size of each PT and LN was also quantified using MIM software and recorded.

### $^{18}$F-FDG-PET/CT analysis and tumor delineation

A nuclear medicine specialist with 10 years of PET experience (R. P.) reviewed all $^{18}$F-FDG-PET/CT images from a dedicated workstation (SyngoVia, Siemens). PET images were analysed both qualitatively (presence/absence of tracer uptake outside sites of physiological accumulation or excretion) and semi-quantitatively. For the latter approach, a volumetric volume of interest (VOI) was placed over the PT and the largest LN. To ensure consistency in the identification of the chosen LN, the delineation was done in consensus with the radiologist. A threshold of 40% $SUV_{max}$ was used to obtain the metabolic tumor volume (MTV), from which $SUV_{max}$, $SUV_{peak}$, $SUV_{mean}$, and the total lesion glycolysis (TLG) were automatically derived. Adjacent FDG-avid structures and areas exhibiting physiological uptake were excluded.

### Statistics

All variables were synthesized through absolute and percentage frequencies and via median values and their relative ranges, when appropriate. Median rather than mean values were used for the analyses, given that the median is less affected by outliers and skewed data. The correlations between all variables were assessed using the Spearman rank correlation test. To exclude false positive results under multiple testing, the Benjamini-Hockberg procedure with a false discovery rate (FDR) of 0.05 was applied.

The paired-sample Wilcoxon signed rank test was used to investigate the differences in imaging parameters between PTs and LNs. The Mann-Whitney test was used to explore the differences between the imaging variables by the HPV status. A $p < 0.05$ was considered statistically significant. The analyses were carried out with SPSS version 21.

### Results

From January 2016 to October 2018 a total of 52 patients affected by OPSCC were retrospectively enrolled in the present study. Patient and tumor characteristics are summarized in Table 1.

Out of 52 patients, 33 were HPV positive and 19 were HPV negative, of whom 13 were graded as G3, 4 as G2 and 2 were without available grading. In 4 patients, evaluation of the PT by DCE-MRI and $^{18}$F-FDG-PET/CT was not possible because the primary lesion was not

**Table 1. Patient and tumor characteristics.**

| Characteristic | | N |
|---|---|---|
| Patients | | 52 |
| Gender | Male | 44 (84.6%) |
| | Female | 8 (15.4%) |
| Age | | 62.32 |
| (years, mean, SD) | | (9.38) |
| Tumor site | Tonsil | 27 |
| | Base of the tongue | 24 |
| | Both | 1 |
| T stage | T1 | 7 |
| | T2 | 11 |
| | T3 | 5 |
| | T4 | 29 |
| N stage | N0 | 3 |
| | N1 | 19 |
| | N2 | 22 |
| | N3 | 8 |
| Primary tumor volume (cm$^3$,SD) | | 18.0 (15.9) |
| Lymph-nodes volume (cm$^3$,SD) | | 11.2 (12.4) |
| Time Interval between MRI and PET-CT (days, SD) | | 16 (15) |
| HPV | + | 33 (63.5%) |
| | - | 19 (36.5%) |

Abbreviations: SD, standard deviation; HPV, human papilloma virus.

visible or too small ($< 0.5$ cm$^3$). In 7 patients, evaluation of the LNs was not feasible because the patients were N0 (3/7), the DCE-MRI did not entirely include the LN (2/7), or the LN was too small (2/7).

Summary statistics of all the variables derived from DCE-MRI and $^{18}$F-FDG-PET/CT are reported in Tables 2 and 3.

PTs showed significantly higher K$^{trans}$ and $v_e$ values, particularly for P10, P25 and P50 percentiles. PTs also showed significantly higher K$_{ep}$ P10, and K$_{ep}$ skewness and kurtosis. At the same time, all $^{18}$F-FDG-PET parameters were larger in PTs than in LNs.

No significant correlation was found between DCE-MRI and $^{18}$F-FDG-PET parameters in PTs (data reported in S1, S2 and S3 Tables), while significant associations emerged between variables derived from K$^{trans}$ and $^{18}$F- FDG-PET in LNs, as shown in Table 4.

Data relative to the correlations between K$_{ep}$/$v_e$ and 18F-FDG-PET parameters in LNs are reported in the S4 and S5 Tables.

In HPV-positive patients, the kurtosis of $v_e$ of PTs was higher than in HPV-negative patients (p = 0.009), while the MTV of LNs was larger (p = 0.025). No other significant difference in imaging parameters by HPV status was found.

Two representative cases are illustrated in Figs 1 and 2.

## Discussion

A multiparametric approach to analyze primary tumors and nodal masses before treatment is advisable in HNSCC, mainly to clarify the complex associations between multiple imaging-based functional biomarkers. These biomarkers have been demonstrated to be useful for

**Table 2. Summary statistics of DCE-MRI parameters in primary tumors (PTs) and metastatic lymph nodes (LNs).**

| Parameter | | PT (N = 48) | | LN (N = 45) | | |
|---|---|---|---|---|---|---|
| | | median | IQR | median | IQR | P |
| $K^{trans}$ | P10 | 0.37 | 0.21 | 0.27 | 0.16 | **0.009** |
| | P25 | 0.53 | 0.28 | 0.42 | 0.19 | **0.012** |
| | P50 | 0.71 | 0.38 | 0.58 | 0.32 | **0.030** |
| | P75 | 0.98 | 0.61 | 0.88 | 0.58 | 0.161 |
| | P90 | 1.27 | 0.92 | 1.17 | 0.96 | 0.595 |
| | Skewness | 2.12 | 1.26 | 2.30 | 1.61 | 0.512 |
| | Kurtosis | 10.88 | 11.27 | 10.71 | 15.48 | 0.442 |
| | Entropy | 4.84 | 0.98 | 4.54 | 1.72 | **0.042** |
| $K_{ep}$ | P10 | 0.88 | 0.40 | 0.72 | 0.50 | **0.002** |
| | P25 | 1.28 | 0.60 | 1.12 | 0.68 | 0.133 |
| | P50 | 1.84 | 0.92 | 1.76 | 1.04 | 0.514 |
| | P75 | 2.64 | 1.68 | 2.48 | 1.92 | 0.677 |
| | P90 | 3.68 | 2.83 | 3.52 | 2.92 | 0.408 |
| | Skewness | 5.64 | 8.75 | 4.48 | 4.77 | **0.032** |
| | Kurtosis | 69.4 | 212 | 40.3 | 86.9 | **0.024** |
| | Entropy | 3.72 | 1.18 | 3.59 | 1.17 | 0.648 |
| $v_e$ | P10 | 0.24 | 0.14 | 0.17 | 0.16 | **0.001** |
| | P25 | 0.33 | 0.15 | 0.24 | 0.13 | **0.002** |
| | P50 | 0.41 | 0.17 | 0.31 | 0.16 | **0.004** |
| | P75 | 0.48 | 0.15 | 0.39 | 0.16 | **0.016** |
| | P90 | 0.54 | 0.17 | 0.49 | 0.21 | 0.051 |
| | Skewness | 0.37 | 0.90 | 0.52 | 0.95 | 0.154 |
| | Kurtosis | 4.55 | 2.19 | 3.86 | 2.64 | 0.253 |
| | Entropy | 4.60 | 0.69 | 4.49 | 0.65 | 0.183 |

Abbreviations: IQR, interquartile range; $K^{trans}$ (min$^{-1}$), transfer constant between plasma and EES (extravascular extracellular space); $K_{ep}$ (min$^{-1}$), transfer constant between EES and plasma; $v_e$, fractional volume of EES; P10, P25, P50, P75, P90 are 10th, 25th, 50th, 75th and 90th percentiles of the volumetric distribution of each parameter inside PT/LN. P values refer to the paired-sample Wilcoxon signed rank test. Statistically significant p-values are **bold.**

differential diagnosis, as well as for predicting and monitoring the treatment response in HNSCC [5,32–35].

**Table 3. Summary statistics of $^{18}$F-FDG-PET parameters in primary tumors (PTs) and metastatic lymph nodes (LNs).**

| Parameter | PT (N = 48) | | LN (N = 49) | | |
|---|---|---|---|---|---|
| | median | IQR | Median | IQR | P |
| $SUV_{max}$ | 17.16 | 9.91 | 10.38 | 8.24 | **<0.001** |
| $SUV_{peak}$ | 13.03 | 6.62 | 6.54 | 6.49 | **<0.001** |
| $SUV_{mean}$ | 10.24 | 5.60 | 6.06 | 4.84 | **<0.001** |
| SD | 2.48 | 1.24 | 1.55 | 1.31 | **<0.001** |
| TLG | 86.76 | 88.68 | 20.18 | 72.47 | **0.005** |
| MTV | 8.49 | 9.02 | 4.95 | 7.33 | **0.034** |

Abbreviations: IQR, interquartile range; $SUV_{max}$, maximum standardized uptake, $SUV_{peak}$ peak standardized uptake within 1 cm$^3$; $SUV_{mean}$, mean standardized uptake; SD, standard deviation of SUV values; TLG, total glycolysis volume; MTV, metabolic tumor volume. P values refer to the paired-sample Wilcoxon signed rank test. Statistically significant p-values are **bold.**

**Table 4. Results of Spearman's correlation tests between K$^{trans}$ and $^{18}$F-FDG-PET parameters in lymph nodes (N = 45).**

| *Variables* | | SUV$_{max}$ | SUV$_{peak}$ | SUV$_{mean}$ | SD | TLG | MTV |
|---|---|---|---|---|---|---|---|
| P10 | Rho | -.375 | -.330 | -.369 | -.305 | -.098 | .001 |
| | P | **.011** | .027 | **.013** | .042 | .524 | .993 |
| P25 | Rho | -.384 | -.346 | -.378 | -.309 | -.168 | -.081 |
| | P | **.009** | .020 | **.011** | .039 | .271 | .599 |
| P50 | Rho | -.405 | -.372 | -.394 | -.331 | -.226 | -.126 |
| | P | **.006** | **.012** | **.007** | .026 | .135 | .408 |
| P75 | Rho | -.429 | -.424 | -.421 | -.374 | -.348 | -.236 |
| | P | **.003** | **.004** | **.004** | **.011** | .019 | .119 |
| P90 | Rho | -.433 | -.443 | -.438 | -.397 | -.402 | -.270 |
| | P | **.003** | **.002** | **.003** | **.007** | **.006** | .073 |
| skewness | Rho | .269 | .221 | .234 | .222 | .123 | .054 |
| | P | .075 | .145 | .122 | .144 | .419 | .724 |
| kurtosis | Rho | .285 | .254 | .260 | .243 | .199 | .110 |
| | P | .057 | .092 | .085 | .107 | .190 | .472 |
| Entropy | Rho | -.296 | -.299 | -.279 | -.234 | -.273 | -.223 |
| | P | .048 | .046 | .063 | .121 | .070 | .141 |

Statistically significant p-values after applying Benjamini-Hockberg correction are **bold** (the corrected p-value threshold is 0.014). Abbreviations as in Tables 2 and 3.

Previous studies have focused on the correlation between perfusion and metabolic imaging in HNSCC, using DCE-MRI and $^{18}$F-FDG PET or $^{18}$F-FMISO (fluoromisonidazole) PET [19–28], as well as with simultaneous PET/MR systems [21,23,32,36]. However, most of these studies investigated perfusion and metabolic parameters in the PT [19–21,23,24,26], while only a few investigations have evaluated the metastatic LNs [25,27,28], reporting conflicting results.

In the present study, we analyzed a homogeneous patient population of OPSCCs, both in PTs and LNs, and found no significant correlation between DCE-MRI and $^{18}$F-FDG-PET in PTs but evident relationships between K$^{trans}$ and $^{18}$F-FDG-PET in LNs.

Prior to performing these analyses, we had explored the potential influence of the HPV status on perfusion and MTV. It is known that HPV-related OPSCC represents a distinct subtype of HNSCC with unique molecular pathogenesis, clinical presentation and prognosis [37]. However, the percentiles of each perfusion parameter, as well as SUV$_{max}$, SUV$_{peak}$, SUV$_{mean}$, did not significantly differ by HPV status. However, the MTV of LNs was found to be higher in the HPV-positive group than in the HPV-negative one. Our results on DCE-MRI are in line with previous investigations that did not report any difference in perfusion parameters according to HPV status, for both PTs and metastatic LNs [22,28]. While conflicting results have been reported on the association between $^{18}$F-FDG-PET/CT parameters and HPV-status, some studies documenting SUV values of PTs have shown that these are lower in HPV-positive than in HPV-negative patients [38–40]. Others have shown no significant difference in the metabolic parameters in nodal metastases [39], as supported by our findings. The lack of significant differences between imaging parameters derived from DCE-MRI and FDG-PET in head and neck cancer by p16 status has also recently been reported by Cao et al. [41].

The larger MTV of LNs in HPV-positive patients may be explained by considering that patients with HPV-related OPSCC are more likely to have a higher N-stage than patients with non-HPV-related OPSCC [38], thus generally exhibiting larger volumes and glycolytic indexes of LNs [39].

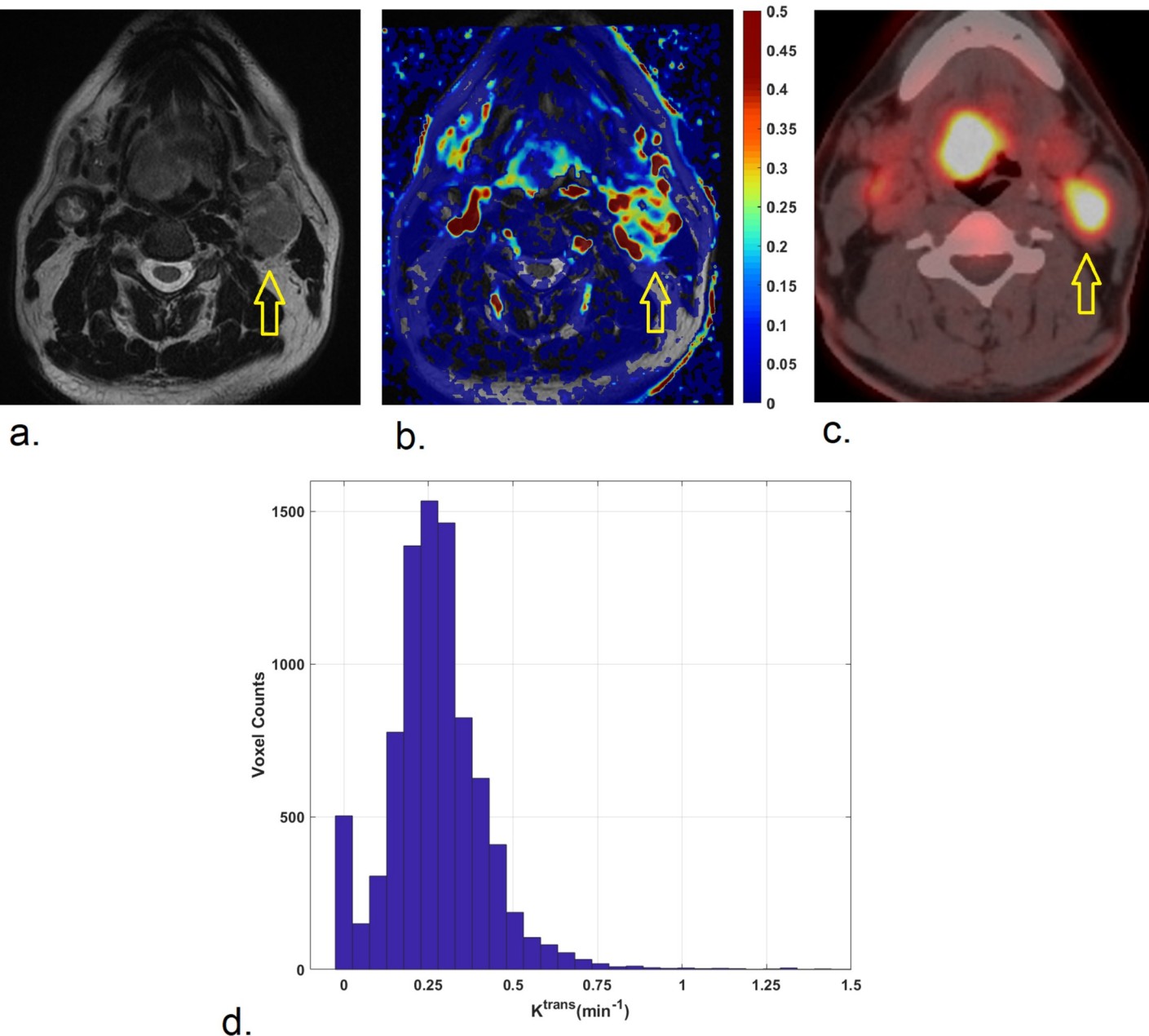

**Fig 1.** 53-year-old man affected by HPV-positive oropharyngeal squamous cell carcinoma of the base of the tongue with a large metastatic lymph-node in the left IIa level, as shown on axial T2-weighted image (a). $K^{trans}$ map (b) indicates heterogeneous $K^{trans}$ levels in the metastatic lymph-node with a low median value of 0.27 $min^{-1}$. Correspondingly, a high $^{18}$F-FDG uptake ($SUV_{max}$: 14.48; $SUV_{peak}$: 10.5; $SUV_{mean}$: 8.76) was found, as illustrated in $^{18}$F-FDG PET/CT image (c). Histogram of $K^{trans}$ values within the entire lymph node is shown (d).

The DCE-MRI and $^{18}$F-FDG-PET parameters of PTs in our study were similar to those reported by Bisdas et al. [24], who investigated the relationships between vascular and metabolic characteristics in primary HNSCC. They found no relationship between $SUV_{max}$/$SUV_{mean}$ and $K^{trans}$/$K_{ep}$, but a significant correlation emerged between $SUV_{mean}$ and $v_e$, which contradicts our present findings. This discrepancy may be attributed to differences in the patient population, as we analyzed a larger and homogenous HNSCC population, in the

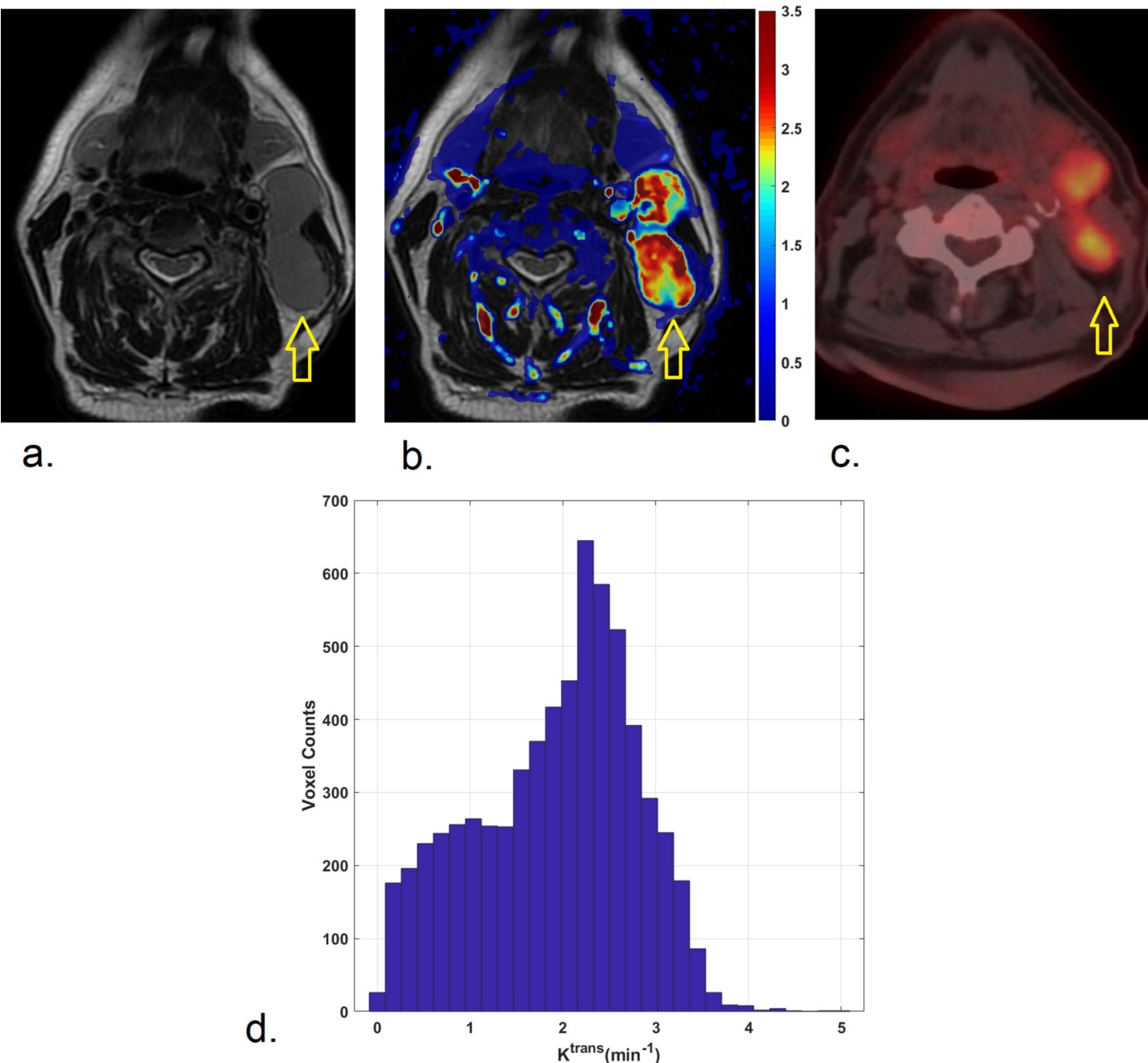

**Fig 2.** 72-year-old man affected by HPV-positive oropharyngeal squamous cell carcinoma of the base of the tongue with enlarged metastatic lymph-nodes in the left IIa/IIb level is shown on axial T2-weighted image (a). $K^{trans}$ map (b) indicates high $K^{trans}$ levels in the lymph-nodes, of which the largest posterior node was analyzed, with a median $K^{trans}$ of 2.08 min$^{-1}$. Correspondingly, a low to intermediate $^{18}$F-FDG uptake ($SUV_{max}$: 6.09; $SUV_{peak}$: 5.82; $SUV_{mean}$: 4.2), was found, as illustrated in $^{18}$F-FDG PET/CT image (c). Histogram of $K^{trans}$ values within the entire lymph node is shown (d).

acquisition protocols of both DCE-MRI and $^{18}$F-FDG-PET/CT, and/or in the methods for image analysis.

As suggested by more recent literature [19], we performed a histogram-based analysis of the DCE-MRI parameters, to consider the vascular heterogeneity within the lesion, and potentially increase the ability to demonstrate associations between perfusion and metabolic

variables. It has already been reported that $SUV_{max}$ is related to $K_{ep}$ P10, and that P25 and TLG tended to be related to $K_{ep}$ P25 and $K^{trans}$ P10 in primary HNSCCs [19]. Moreover, it has been suggested that the evidence of these correlations may also depend on tumor grading, with G1/G2 PTs showing significant associations, while no correlations are evident in G3 PTs [19, 13]. This may partially explain the lack of correlations in PTs emerging from our study: in our cohort, HPV-negative OPSCCs were predominately G3 (13/19 patients) while no grading system currently exists for HPV-positive OPSCCs according to the American Joint Committee on Cancer Staging Manual [30].

It should also be stressed that, unlike other investigators, we applied a p-value correction to exclude false positive results under multiple testing, and this may have contributed to reducing the evidence of correlations, as well as corroborating our findings.

Concerning the LNs, all $K^{trans}$ percentiles showed strong associations with SUV values, with $K^{trans}$ P90 also correlating with TLG. Our data suggest that the complex relationships between perfusion and metabolic biomarkers should be interpreted separately for PTs and LNs. This may be attributed to the differences in tissue microvascular architecture between the PT and the pathological lymphadenopathy [42]. This difference between PTs and LNs is also compounded by the fact that LNs had significantly lower vascular and metabolic values than PTs. This is in line with Fischbein et al. [42], who observed that semi-quantitative perfusion parameters of LNs, as peak enhancement and maximum slope of signal increase, were unexpectedly lower in tumor-involved compared with non-tumor- involved LNs. This may suggest that, especially for reactive nodal tissue, the tumor does not necessarily show higher vascularity compared with normal lymphoid tissue.

Recently, possible associations between 18F-FDG-PET and microvessel density (MVD) have been evaluated in HNSCC [43]. MVD assessments have been proposed as measures of tumor vascularity, based on the expression levels of some vascular endothelium markers by immunohistochemistry [44]. Surov et al. [43] found that $SUV_{max}$ correlated with vessel area and vessel count in PTs. Unfortunately, there are no reports of similar analyses in malignant cervical LNs, which could have been helpful in explaining our findings.

Previous studies have also investigated the relationship between DCE-MRI and [18]F-FMISO PET in HN neck nodal metastases [25], showing that hypoxic nodes are poorly perfused compared to nodes without hypoxia with a negative correlation between FMISO uptake and the median $K_{ep}$ value. At the same time, positive correlations were observed between FMISO uptake and FDG uptake in LNs [25], and between hypoxic volume using [18]F-FMISO and hypermetabolic volume using [18]F-FDG in HN cancer [27], suggesting that the presence of hypoxia may lead to a greater glucose uptake. The above-mentioned considerations may help explain our findings, even though further studies are needed to better clarify the complex interplay between multi-modal imaging measurements.

To this aim, it would be of interest to evaluate the associations between [18]F-FDG PET and ADC measurements for a better tumor characterization, as proposed by some investigators [45,46]. The findings were highly incongruent, showing either no significant correlations or a wide range of correlation coefficients between FDG-PET parameters and ADC [45,46], with a possible dependence on the tumor grade [46]. Interestingly, Teng et al [47] also investigated the spatial relationship between tumor subvolumes of high FDG uptake, low blood volume, and low ADC values in HN cancer, suggesting that multiple imaging techniques, instead of a single imaging modality, should be used to define a potential boosting target and adequately identify tumor subvolumes at higher risk of treatment failure.

There were some limitations in the current study. First, its retrospective nature may have introduced bias and confounding factors. This also prevented us from performing a correlation study at the voxel level, which would have required an accurate image co-registration

between PET and MR studies, using a similar patient positioning in both scans. The histogram analysis was proposed only for DCE-MRI maps, and not for $^{18}$F-FDG PET images, considering the large difference in spatial resolution between the two imaging modalities. We could not have explored the influence of the tumor grading on the strength of the associations between imaging parameters, as our population had a larger proportion of high-grade tumors.

In conclusion, evident relationships emerged between DCE-MRI and $^{18}$F-FDG PET parameters in OPSCC LNs, while no association was found in PTs. Further studies are warranted for a better understanding of the underlying interactions between microvascular properties and tumor metabolism in both tumor sites. These studies would support both radiologists and clinicians in identifying which parameters, alone or in combination, should be proposed in clinical practice for more precise diagnosis and personalized treatment protocols.

## Supporting information

**S1 Table. Results of Spearman's correlation tests between $K^{trans}$ and $^{18}$F-FDG-PET parameters in primary tumors (N = 47).**
(DOCX)

**S2 Table. Results of Spearman's correlation tests between $K_{ep}$ and $^{18}$F-FDG-PET parameters in primary tumors (N = 47).**
(DOCX)

**S3 Table. Results of Spearman's correlation tests between $v_e$ and $^{18}$F-FDG-PET parameters in primary tumors (N = 47).**
(DOCX)

**S4 Table. Results of Spearman's correlation tests between $K_{ep}$ and $^{18}$F-FDG-PET parameters in lymph nodes (N = 45).**
(DOCX)

**S5 Table. Results of Spearman's correlation tests between $v_e$ and $^{18}$F-FDG-PET parameters in lymph nodes (N = 45).**
(DOCX)

**S1 Data.**
(XLSX)

## Author Contributions

**Conceptualization:** Antonello Vidiri, Giuseppe Sanguineti, Simona Marzi.

**Data curation:** Emma Gangemi, Emanuela Ruberto, Alessia Farneti, Maria Benevolo, Francesca Rollo, Francesca Sperati, Filomena Spasiano, Raul Pellini.

**Formal analysis:** Emma Gangemi, Emanuela Ruberto, Rosella Pasqualoni, Rosa Sciuto, Alessia Farneti, Francesca Rollo, Francesca Sperati, Filomena Spasiano, Simona Marzi.

**Funding acquisition:** Antonello Vidiri, Giuseppe Sanguineti.

**Investigation:** Emma Gangemi, Emanuela Ruberto, Alessia Farneti, Maria Benevolo, Francesca Sperati, Raul Pellini, Simona Marzi.

**Methodology:** Antonello Vidiri, Emma Gangemi, Emanuela Ruberto, Rosella Pasqualoni, Rosa Sciuto, Giuseppe Sanguineti, Alessia Farneti, Maria Benevolo, Francesca Sperati, Simona Marzi.

**Project administration:** Simona Marzi.

**Resources:** Rosella Pasqualoni, Rosa Sciuto.

**Software:** Rosella Pasqualoni, Rosa Sciuto, Simona Marzi.

**Supervision:** Antonello Vidiri, Giuseppe Sanguineti.

**Validation:** Antonello Vidiri, Giuseppe Sanguineti, Francesca Sperati, Simona Marzi.

**Visualization:** Emma Gangemi, Emanuela Ruberto, Rosella Pasqualoni, Rosa Sciuto, Alessia Farneti, Maria Benevolo, Francesca Rollo, Raul Pellini.

**Writing – original draft:** Antonello Vidiri, Emma Gangemi, Rosa Sciuto, Giuseppe Sanguineti, Maria Benevolo, Francesca Sperati, Simona Marzi.

**Writing – review & editing:** Antonello Vidiri, Emma Gangemi, Simona Marzi.

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
