## [Decision Letter · Decision Letter 0]

11 Dec 2019

PONE-D-19-29908

Correlation between histogram-based DCE-MRI parameters  and 18F-FDG PET values in oropharyngeal squamous cell carcinoma: Evaluation in primary tumors and metastatic nodes

PLOS ONE

Dear Dr gangemi,

Thank you for submitting your manuscript to PLOS ONE. After careful consideration, we feel that it has merit but does not fully meet PLOS ONE’s publication criteria as it currently stands. Therefore, we invite you to submit a revised version of the manuscript that addresses the points raised during the review process.

Both the reviewers and myself had some concerns regarding the language both in terms of grammatical errors and clarity. I suggest that you have the manuscript revised by a native English speaker to improve the overall quality of the work. Please note that PLOS ONE does minimal copy-editing upon acceptance of a manuscript, so it is important that the final version does not have grammar and spelling mistakes.

Also, I would also like to invite you to the carefully check that your manuscript is in line with the STROBE reporting requirements for retrospective studies. You can find an associated checklist here: https://www.strobe-statement.org/index.php?id=available-checklists 

Please ensure that all items are met as several are currently missing (e.g. Describe the setting, locations, and relevant dates, including periods of recruitment, exposure, follow-up, and data collection; Explain how the study size was arrived at).

We would appreciate receiving your revised manuscript by Jan 25 2020 11:59PM. To enhance the reproducibility of your results, we recommend that if applicable you deposit your laboratory protocols in protocols.io, where a protocol can be assigned its own identifier (DOI) such that it can be cited independently in the future. For instructions see: http://journals.plos.org/plosone/s/submission-guidelines#loc-laboratory-protocols

We look forward to receiving your revised manuscript.

Kind regards,

Niels Bergsland

Academic Editor

PLOS ONE

Journal Requirements:

"The present study was authorized by the hospital ethics committee".    

a.Please amend your current ethics statement to include the full name of the ethics committee/institutional review board(s) that specifically approved your study.  

b.Once you have amended this/these statement(s) in the Methods section of the manuscript, please add the same text to the “Ethics Statement” field of the submission form (via “Edit Submission”).

3. In the ethics statement in the manuscript and in the online submission form, please provide additional information about the patient records used in your retrospective study, including:

a) whether all data were fully anonymized before you accessed them;

b) the date range (month and year) during which patients' tissue samples and medical records were accessed;

c) the date range (month and year) during which patients whose tissue samples and medical records were selected for this study sought treatment; and

d) the source of the tissue samples and medical records analyzed in this work (e.g. hospital, institution or medical center name).

4. We noticed minor instances of text overlap with the following previous publication(s), which need to be addressed:

(1) https://linkinghub.elsevier.com/retrieve/pii/S0720048X19302839

(2) https://journals.sagepub.com/doi/10.1177/0284185119862946

The text that needs to be addressed involves the (1) Abstract and (1,2) the Methods section.

In your revision please ensure you cite all your sources (including your own works), and quote or rephrase any duplicated text. Further consideration is dependent on these concerns being addressed.

5. Please include captions for your Supporting Information files at the end of your manuscript, and update any in-text citations to match accordingly. Please see our Supporting Information guidelines for more information: http://journals.plos.org/plosone/s/supporting-information

Reviewers' comments:

Reviewer's Responses to Questions

**Comments to the Author**

1. Is the manuscript technically sound, and do the data support the conclusions?

Reviewer #1: Yes

Reviewer #2: Yes

2. Has the statistical analysis been performed appropriately and rigorously? 

Reviewer #1: Yes

Reviewer #2: Yes

3. Have the authors made all data underlying the findings in their manuscript fully available?

Reviewer #1: Yes

Reviewer #2: Yes

4. Is the manuscript presented in an intelligible fashion and written in standard English?

Reviewer #1: Yes

Reviewer #2: Yes

5. Review Comments to the Author

Reviewer #1: Correlation between histogram-based DCE-MRI parameters and 18F-FDG PET

values in oropharyngeal squamous cell carcinoma: Evaluation in primary tumors and

metastatic nodes

MAJOR Weakness

Unclear clinical relevance of the study in the present form

Abstract

OK

Key words: OK

Introduction

In my point of view, in the introduction, data about associations between imaging findings and histopathology in different malignancies should be given.

It is well known that besides clinical and histopathological factors, imaging parameters, especially those derived from PET, DWI and DCE MRI are important prognostic biomarkers in different malignancies.

The purpose of the study, especially, clinical relevance etc. is unclear defined.

M&M

OK. Please see my suggestions for the section results.

Results

Interesting data.

In my point of view, you should also analyse associations between DCE MRI and PET parameters in HPV+ and HPV- tumors separately.

Discussion

Well.

However, there are reports regarding relationships between PET and microvessel density in different tumors, also in HNSCC. These associations may explain correlations between DCE MRI and PET parameters.

Please discuss

Conclusion:

Please give more detailed possible clinical relevance of your results.

References:

Some references may be added (see my suggestions above).

Figures

Well

Tables

Well

Reviewer #2: This is an interesting article, which studies the intercorrelation between standard FDG PET and DCE MRI imaging in a limited head and neck cancer patient population. While the study is scientifically sound, there are a number of issues that should be addressed prior to publication. Most of these issues are grammatical in nature, but some may influence the overall conclusions of the work. The work has not been published before to my knowledge.

In general the manuscript needs full English-language review. For example, first two sentences in Discussion need significant work (I could not ascertain what the authors were trying to say), as do Lines 353+. Line 384 "while no tendency of correlations..." is not a standard use of English. Full English-language review is recommended prior to publication.

Specific Comments (including science questions):

FDG Tumor Delineation:

Was the largest LN on PET always the same largest LN on MRI? How was it ensured that the same LN was being analyzed on both imaging modalities. If these are not the same LN this could have a profound effect on the overall conclusions of the paper.

Table 2 and 3: This is a very confusing header row. We have PT (Primary Tumor) with median and IQR, then we have LN (Lymph Nodes) with median and IQR. Then we have a column "Both" but only a P value (not a median or IQR). I would assume you would want the DCE/PET parameters in the PT and LNs separately in addition to the combined Tumor (PT+LNs) but this doesn't seem to be the case? The headings here are confusing.

Please comment on ADC metrics and refer to recent paper looking at ADC vs. FDG/PET in H/N cancer: https://www.frontiersin.org/articles/10.3389/fonc.2019.01118/full which showed high correlations between MRI and FDG metrics in a very similar (and large) patient population. Although impact is not directly assessed in this review, this paper should be referenced since it showed that correlation between high glucose metabolism and high restricted water diffusion varied greatly spatially from patient to patient.

Line 72: correct LSs to LNs

Line 115: even "though" not even "if"

Line 217: Were scans attenuation corrected? If so, please include this in addition to all corrections (Time-of-Flight? Decay?)

Line 401: Period is needed.

6. PLOS authors have the option to publish the peer review history of their article (what does this mean?). If published, this will include your full peer review and any attached files.

Reviewer #1: No

Reviewer #2: Yes: Benjamin Rosen

---

## [Author Response · Author response to Decision Letter 0]

29 Jan 2020

In response to the Editor:

1) Both the reviewers and myself had some concerns regarding the language both in terms of grammatical errors and clarity. I suggest that you have the manuscript revised by a native English speaker to improve the overall quality of the work. Please note that PLOS ONE does minimal copy-editing upon acceptance of a manuscript, so it is important that the final version does not have grammar and spelling mistakes.

R: As suggested, a full English-language revision has been performed prior to submit the revised manuscript.

2) Also, I would also like to invite you to the carefully check that your manuscript is in line with the STROBE reporting requirements for retrospective studies. You can find an associated checklist here: https://www.strobe-statement.org/index.php?id=available-checklists

Please ensure that all items are met as several are currently missing (e.g. Describe the setting, locations, and relevant dates, including periods of recruitment, exposure, follow-up, and data collection; Explain how the study size was arrived at).

R: We have checked that the manuscript is in line with the STROBE reporting requirements for retrospective studies. In particular:

• we have reported in the abstract the following sentence: “52 patients with a pathologically confirmed OPSCC were included in the present retrospective cohort study” (line 59-60);

• we have added the setting in the methods of the manuscript: “This cohort study was conducted retrospectively at the IRCCS Regina Elena National Cancer Institute, Rome, Italy” (line 135-136);

• we have added the timing in the methods (All patients' tissue samples and medical records were accessed between November 2018 and April 2019, line 156-157) and results (From January 2016 to October 2018 a total of 52 patients affected by OPSCC were retrospectively enrolled in the present study, line 262-263).

• Time of Follow up is not included in the manuscript, because it is not relevant for the aim of the study.

• Due to the retrospective nature of the study and the lack of a specific clinical endpoint, we decided to consider adequate a sample size of about 50 patients, based on the number of patients coming into our institute in the observational period selected. In fact, considering the rarity of the disease and the advanced methodologies used, no published data could have supported specific hypothesis for a conventional sample size calculation (line 143-147). 

3) To enhance the reproducibility of your results, we recommend that if applicable you deposit your laboratory protocols in protocols.io, where a protocol can be assigned its own identifier (DOI) such that it can be cited independently in the future. For instructions see: http://journals.plos.org/plosone/s/submission-guidelines#loc-laboratory-protocols.

R: The deposit of our laboratory protocols in protocols.io is not applicable.

In response to Reviewer #1: 

MAJOR Weakness

1) Unclear clinical relevance of the study in the present form

R: We thank the Reviewer for this comment. The clinical relevance of the study has been better addressed in introduction (line 100-109) and Discussion (line 334-338; 438-441).

2) Abstract OK

3) Key words: OK

4) Introduction. 

In my point of view, in the introduction, data about associations between imaging findings and histopathology in different malignancies should be given. It is well known that besides clinical and histopathological factors, imaging parameters, especially those derived from PET, DWI and DCE MRI are important prognostic biomarkers in different malignancies. The purpose of the study, especially, clinical relevance etc. is unclear defined.

R: Thank you for this suggestion. We have modified the introduction, giving more emphasis on the associations between imaging findings and histopathology, also in different malignancies (line 102-114). To address this point, we have added some new references, which are indicated below. The purpose of the study related to its potential clinical relevance is now better defined.

New references:

1. Surov A, Meyer HJ, Wienke (2018) A. Can Imaging Parameters Provide Information Regarding Histopathology in Head and Neck Squamous Cell Carcinoma? A Meta-Analysis. Transl Oncol;11(2):498-503. doi: 10.1016/j.tranon.2018.02.004. 

2. Zheng H, Cui Y et al. (2019) Prognostic Significance of 18F-FDG PET/CT Metabolic Parameters and Tumor Galectin-1 Expression in Patients With Surgically Resected Lung Adenocarcinoma Clin Lung Cancer 20(6):420-428. doi: 10.1016/j.cllc.2019.04.002. 

3. Incoronato M, Grimaldi AM, Cavaliere C, et al.(2018) Relationship between functional imaging and immunohistochemical markers and prediction of breast cancer subtype: a PET/MRI study. Eur J Nucl Med Mol Imaging. 2018 Sep;45(10):1680-1693. doi: 10.1007/s00259-018-4010-7. 

4. Bekaert L, Valable S, Lechapt-Zalcman E et al. (2017) [18F]-FMISO PET study of hypoxia in gliomas before surgery: correlation with molecular markers of hypoxia and angiogenesis. Eur J Nucl Med Mol Imaging.;44(8):1383-1392. doi: 10.1007/s00259-017-3677-5. 

5. Rasmussen GB, Vogelius IR, Rasmussen JH, et al. (2015) Immunohistochemical biomarkers and FDG uptake on PET/CT in head and neck squamous cell carcinoma. Acta Oncol.;54(9):1408-15. doi: 10.3109/0284186X.2015.1062539. 

6. Unetsubo T1, Konouchi H, Yanagi Y, et al. (2009) Dynamic contrast-enhanced magnetic resonance imaging for estimating tumor proliferation and microvessel density of oral squamous cell carcinomas. Oral Oncol. 2009 Jul;45(7):621-6. doi: 10.1016/j.oraloncology.2008.09.003.

5) M&M see my suggestions for the section results.

R: As suggested for the section results, we have performed the correlation tests in HPV+ and HPV- tumors separately, although we have preferred not to include these analyses in the manuscript for the reasons explained below.

6) Results. 

You should also analyze associations between DCE MRI and PET parameters in HPV+ and HPV- tumors separately.

R: We had explored and discussed the potential influence of the HPV status on imaging parameters derived from DCE-MRI and FDG-PET, as reported in the original version of the manuscript (Statistics: line 260-261; Results: line 313-315; Discussion: line 353-369). Because DCE-MRI parameters, as well as SUVmax, SUVpeak, SUVmean did not significantly differ by the HPV status (based on the Mann-Whitney test results), we decided not to stratify by HPV, also in order not to reduce the statistical power of the analyses: our population had a larger proportion of HPV positive (N = 33) than HPV negative (N=19), thus the statistical power would be limited by the small sample size, particularly in HPV negative patients. 

The lack of significant differences between imaging parameters derived from DCE-MRI and FDG-PET in head and neck cancer by p16 status has also recently been reported by Cao et al. [41]. (Cao Y, Aryal M, Li P et al. (2019) Predictive Values of MRI and PET Derived Quantitative Parameters for Patterns of Failure in Both p16+ and p16- High Risk Head and Neck Cancer. Front Oncol 9:1118. doi: 10.3389/fonc.2019.01118.), which has been added in the reference list.

However, as suggested, we have performed the correlation tests in HPV+ and HPV- tumors separately, although we have preferred not to include these analyses in the manuscript for the reasons mentioned above. The results are illustrated in the Tables inserted at the end of the attached document called "Response to Reviewers": no correlation reached statistically significance after correction for multiple testing in either group. 

7) Discussion. 

There are reports regarding relationships between PET and microvessel density in different tumors, also in HNSCC. These associations may explain correlations between DCE MRI and PET parameters.

Please discuss

R: Thank you for this suggestion. We have now included some reports evaluating the relationships between PET and microvessel density in HNSCC in Discussion. Unfortunately, there are no reports addressing these analyses in malignant cervical lymph nodes, which could have been helpful in explaining our findings.

 A paragraph has been added, as follows (line 400-406):

Recently, possible associations between 18F-FDG-PET and microvessel density (MVD) have been evaluated in HNSCC [43]. MVD assessments have been proposed as measures of tumor vascularity, based on the expression levels of some vascular endothelium markers by immunohistochemistry [44]. Surov et al. found that SUVmax correlated with vessel area and vessel count in PTs. Unfortunately, there are no reports of similar analyses in malignant cervical LNs, which could have been helpful in explaining our findings.

References added: 

1. Szafarowski T, Sierdzinski J, Szczepanski MJ, Whiteside TL, Ludwig N, Krzeski A (2018) Microvessel density in head and neck squamous cell carcinoma. Eur Arch Otorhinolaryngol 275:1845-1851. Doi: 10.1007/s00405-018-4996-2.

2. Surov A, Meyer HJ, Höhn AK, Wienke A, Sabri O, Purz S (2019) 18F-FDG-PET Can Predict Microvessel Density in Head and Neck Squamous Cell Carcinoma. Cancers 11(4). pii: E543. DOI: 10.3390/cancers11040543.

Due to the increased length of the manuscript and the increased number of references, as a consequence of the revision process, we did not mention papers focused on the relationships between PET and microvessel in other malignancies.

8) Conclusion:

Please give more detailed possible clinical relevance of your results.

R: We thank the reviewer for this comment, we have now better emphasized the possible clinical relevance of our results throughout the Discussion and in Conclusion (line 334-338; 438-441)

9) References:

Some references may be added (see my suggestions above).

R: As suggested above, we have now added new references regarding both the associations between imaging findings and histopathology and the relationships between PET and microvessel density.

In response to Reviewer #2: 

1) In general the manuscript needs full English-language review. For example, first two sentences in Discussion need significant work (I could not ascertain what the authors were trying to say), as do Lines 353+. Line 384 "while no tendency of correlations..." is not a standard use of English. Specific Comments (including science questions):

R: As suggested, a full English-language revision has been performed prior to submit the revised manuscript.

2) FDG Tumor Delineation:

Was the largest LN on PET always the same largest LN on MRI? How was it ensured that the same LN was being analyzed on both imaging modalities. If these are not the same LN this could have a profound effect on the overall conclusions of the paper.

R: Yes, the largest LN on PET was always the same largest LN on MRI. The contouring on MRI and PET-CT was performed in consensus by the radiologist and the nuclear doctor. This is now specified in the text (line 242-244 “To ensure consistency in the identification of the chosen LN, the delineation was done in consensus with the radiologist”).

3) Table 2 and 3: This is a very confusing header row. We have PT (Primary Tumor) with median and IQR, then we have LN (Lymph Nodes) with median and IQR. Then we have a column "Both" but only a P value (not a median or IQR). I would assume you would want the DCE/PET parameters in the PT and LNs separately in addition to the combined Tumor (PT+LNs) but this doesn't seem to be the case? The headings here are confusing.

R: In the header of Table 2 and 3, the term ‘Both’ and the number in brackets specifies the number of patients having both a PT and a LN analyzed, in fact some patients do not have the PT analyzed, while some others do not have a LN analyzed as described in Results (line 269-274). However, because it may appear confusing, the term ‘Both’ has been removed from the header of Table 2 and 3.

4) Please comment on ADC metrics and refer to recent paper looking at ADC vs. FDG/PET in H/N cancer: https://www.frontiersin.org/articles/10.3389/fonc.2019.01118/full which showed high correlations between MRI and FDG metrics in a very similar (and large) patient population. Although impact is not directly assessed in this review, this paper should be referenced since it showed that correlation between high glucose metabolism and high restricted water diffusion varied greatly spatially from patient to patient.

R: As suggested, we have read the study of Cao Y et al (https:// www. frontiersin.org/articles/10.3389/fonc.2019.01118/full) and have mentioned it in Discussion (line 360-362) when addressing the potential influence of the HPV status on imaging parameters. In fact, in this paper the authors did not report correlation analyses between parameters derived from MRI and FDG metrics or between their subvolumes but they compared the predictive power of MRI and PET biomarkers in terms of local, regional of distant failure after chemoradiation in advanced head and neck cancer patients, including the HPV status. 

With regard to the correlation between high glucose metabolism and high restricted water diffusion, we have found more relevant the paper of Teng et al.(Teng F, Aryal M, Lee J et al. (2018) Adaptive Boost Target Definition in High-Risk Head and Neck Cancer Based on Multi-imaging Risk Biomarkers. Int J Radiat Oncol Biol Phys.102:969-977. doi: 10.1016/j.ijrobp.2017.12.269). In this study, the spatial relationship between FDG uptake, perfusion and ADC in HN cancer patients were investigated, to evaluate potential implication of their spatial overlap for adaptive boosting of radiotherapy. We found this paper very interesting, suggesting that multiple imaging techniques, instead of a single imaging modality, should be used to define a potential boosting target and adequately identify tumor subvolume at higher risk of treatment failure. We mentioned it in Discussion (line 416-425) as follows:

“…To this aim, it would be of interest to evaluate the associations between 18F-FDG PET and ADC measurements for a better tumor characterization, as proposed by some investigators [45,46]. The findings were highly incongruent, showing either no significant correlations or a wide range of correlation coefficients between FDG-PET parameters and ADC [45,46], with a possible dependence on the tumor grade [46]. Interestingly, Teng et al [47] also investigated the spatial relationship between tumor subvolumes of high FDG uptake, low blood volume, and low ADC values in HN cancer, suggesting that multiple imaging techniques, instead of a single imaging modality, should be used to define a potential boosting target and adequately identify tumor subvolumes at higher risk of treatment failure.”

Reference added:

• Cao Y, Aryal M, Li P et al. (2019) Predictive Values of MRI and PET Derived Quantitative Parameters for Patterns of Failure in Both p16+ and p16- High Risk Head and Neck Cancer. Front Oncol. 9:1118. DOI: 10.3389/fonc.2019.01118.

• Teng F, Aryal M, Lee J et al. (2018) Adaptive Boost Target Definition in High-Risk Head and Neck Cancer Based on Multi-imaging Risk Biomarkers. Int J Radiat Oncol Biol Phys.102:969-977. doi: 10.1016/j.ijrobp.2017.12.269.

• Meyer HJ, Purz S, Sabri O, Surov A. (2018) Relationships between histogram analysis of ADC values and complex 18F-FDG-PET parameters in head and neck squamous cell carcinoma. PLoS One ;13:e0202897. doi: 10.1371/journal.pone.0202897.

• Shen G, Ma H, Liu B, Ren P, Kuang A. (2017) Correlation of the apparent diffusion coefficient and the standardized uptake value in neoplastic lesions: a meta-analysis. Nucl Med Commun. 2017;38:1076-1084. DOI: 10.1097/MNM.0000000000000746.

5) Line 72: correct LSs to LNs 

R: DONE

6) Line 115: even "though" not even "if" 

R: DONE

7) Line 217: Were scans attenuation corrected? If so, please include this in addition to all corrections (Time-of-Flight? Decay?)

R: Our PET/CT scanner has not Time-of-Flight capability. PET images were corrected for attenuation using data from CT. This is now specified in the text (Materials and Methods, 18F-FDG-PET /CT image acquisition paragraph, line 201; 209-210).

8) Line 401: Period is needed. 

R: DONE

---

## [Decision Letter · Decision Letter 1]

11 Feb 2020

Correlation between histogram-based DCE-MRI parameters  and 18F-FDG PET values in oropharyngeal squamous cell carcinoma: Evaluation in primary tumors and metastatic nodes

PONE-D-19-29908R1

Dear Dr. gangemi,

We are pleased to inform you that your manuscript has been judged scientifically suitable for publication and will be formally accepted for publication once it complies with all outstanding technical requirements.

With kind regards,

Niels Bergsland

Academic Editor

PLOS ONE

Additional Editor Comments (optional):

Reviewers' comments:

Reviewer's Responses to Questions

**Comments to the Author**

1. If the authors have adequately addressed your comments raised in a previous round of review and you feel that this manuscript is now acceptable for publication, you may indicate that here to bypass the “Comments to the Author” section, enter your conflict of interest statement in the “Confidential to Editor” section, and submit your "Accept" recommendation.

Reviewer #1: All comments have been addressed

Reviewer #2: All comments have been addressed

2. Is the manuscript technically sound, and do the data support the conclusions?

Reviewer #1: Yes

Reviewer #2: Yes

3. Has the statistical analysis been performed appropriately and rigorously? 

Reviewer #1: Yes

Reviewer #2: Yes

4. Have the authors made all data underlying the findings in their manuscript fully available?

Reviewer #1: Yes

Reviewer #2: No

5. Is the manuscript presented in an intelligible fashion and written in standard English?

Reviewer #1: Yes

Reviewer #2: Yes

6. Review Comments to the Author

Reviewer #1: The authors addressed my suggestions adequately. The manuscript in the present form provides interesting data About associations between DCE MRI and PET Parameters in HNSCC (HPV+ and HPV- Tumors).

Reviewer #2: Comments have been adequately addressed. Manuscript is clearer now and relevant references have been incorporated. Thank you for the opportunity to review this work.

7. PLOS authors have the option to publish the peer review history of their article (what does this mean?). If published, this will include your full peer review and any attached files.

Reviewer #1: No

Reviewer #2: No

---

## [Editor Report · Acceptance letter]

14 Feb 2020

PONE-D-19-29908R1 

Correlation between histogram-based DCE-MRI parameters and ^18^F-FDG PET values in oropharyngeal squamous cell carcinoma: Evaluation in primary tumors and metastatic nodes 

Dear Dr. gangemi:

I am pleased to inform you that your manuscript has been deemed suitable for publication in PLOS ONE. Congratulations! Your manuscript is now with our production department. 

With kind regards,

on behalf of

Dr. Niels Bergsland 

Academic Editor

PLOS ONE